# BOND: Bootstrapping From-Scratch Name Disambiguation with Multi-task Promoting

## ABSTRACT

From-scratch name disambiguation is an essential task for establishing a reliable foundation for academic platforms. It involves partitioning documents authored by identically named individuals into groups representing distinct real-life experts. Canonically, the process is divided into two decoupled tasks: locally estimating the pairwise similarities between documents followed by globally grouping these documents into appropriate clusters. However, such a decoupled approach often inhibits optimal information exchange between these intertwined tasks. Therefore, we present BOND, which bootstraps the local and global informative signals to promote each other in an end-to-end regime. Specifically, BOND harnesses local pairwise similarities to drive global clustering, subsequently generating pseudo-clustering labels. These global signals further refine local pairwise characterizations. The experimental results establish BOND's superiority, outperforming other advanced baselines by a substantial margin. Moreover, an enhanced version, BOND+, incorporating ensemble and post-match techniques, rivals the top methods in the WhoIsWho competition[1].

## KEYWORDS

name disambiguation, multi-task learning

**ACM Reference Format:**
Anonymous Author(s). 2024. BOND: Bootstrapping From-Scratch Name Disambiguation with Multi-task Promoting. In *Proceedings of the ACM Web Conference 2024 (WWW '24)*. ACM, New York, NY, USA, 10 pages. https://doi.org/XXXXXXX.XXXXXXX

**Relevance to the Web and to the track.** Author name disambiguation is increasingly complex due to the surge in online publications. These papers originate from various platforms like Web of Science and Google Scholar. Precise name disambiguation is vital for accurate academic search and user query responses. This process is closely tied to *Web mining and content analysis*, essential for integrating diverse online publications and ensuring data quality.

## 1 INTRODUCTION

Name disambiguation is a core component in online academic systems such as Google Scholar, DBLP, and AMiner [35]. With the exponential growth of research documents in recent years [44],

[1]http://whoiswho.biendata.xyz/

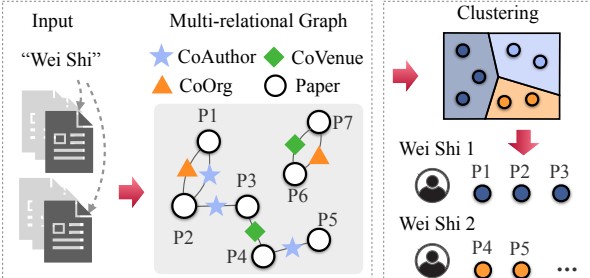

(a) Overview of the SND problem

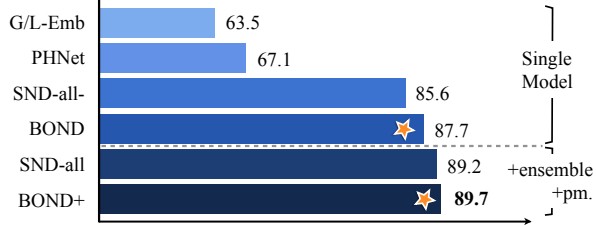

(b) Performance comparison on the WhoIsWho leaderboard (F1%)

**Figure 1: An overview of the SND problem and performance comparisons between BOND and baselines.** (a) Paper connections are established through diverse relationships. Noise is observed in the linkage of Paper $p4$ to Paper $p3$; (b) *pm.*: post-match.

the problem of author name ambiguity has become more complex. This issue encompasses scenarios where identical authors exhibit diverse name variations, distinct authors share identical names, or instances of homonyms. For instance, as of October 2023, DBLP contained over 300 author profiles with the name "Wei Wang" in the field of computer science alone, not to mention across all academic disciplines. This underscores the pressing demand for the development of efficient and scalable algorithms tailored to confront the challenges presented by author name ambiguity.

In this paper, we delve into the important task of From-Scratch Name Disambiguation (SND), which is fundamental for building digital libraries. The main goal, as shown in Figure 1 (a), is to organize papers linked to the same author's name into separate author profiles, each representing an individual's work. However, due to the missing, fragmented, and noisy paper attributes (e.g. author email, author organizations) across data sources, the performance of SND methods are still unsatisfactory. Previous research has traditionally treated SND as a clustering problem, which can be broken down into two main tasks: (1). *Local Metric Learning.* This task concentrates on assessing fine-grained similarities among papers. It typically uses advanced embedding techniques to transform these papers into lower-dimensional representations. Then, metric functions are applied to calculate local pairwise similarities among these papers. (2). *Global Clustering.* With the learned local relationship of these papers, clustering methods are usually used to acquire the

global partition of these papers, where the papers owned by the same author are divided into the same group.

Unfortunately, previous methods often approached these two stages as two successive decoupled phases. To clarify, early attempts [1, 14, 21, 37] employed hand-crafted pairwise paper similarity features, in conjunction with traditional classifiers such as SVM [12], to establish similarity metric functions. Then, during the global clustering phase, algorithms like DBSCAN [8] were used to group papers into distinct clusters. Recent approaches have ventured into building homogeneous paper similarity graphs [19, 46] based on co-author or other relationships, or constructing heterogeneous graphs [29, 30] to capture high-order connections. For example, PHNet [29] leverages heterogeneous network embedding techniques to obtain paper representations and employs sophisticated clustering methods to categorize papers into clusters. However, this isolated learning approach faces challenges in effectively combining the information from local pairwise metric learning and global clustering signals. This separation may result in accumulating errors that are difficult to correct during the training process.

**Present Work.** Building upon the insights mentioned above, we present BOND, a BOotstrapping From-Scratch Name Disambiguation with Multi-task Promoting approach, to bootstrap the local and global informative signals to each other in an end-to-end regime.

Specifically, BOND consists of three key components: 1). *Multi-relational Graph Construction*. BOND carefully devises strategies for constructing graphs, ensuring the preservation of multi-relational connections among paper nodes. 2). *Local Metric Learning via Edge Reconstruction*. Leveraging a graph auto-encoder with the Graph Attention Network (GAT) [39] as the encoder, BOND learns paper representations via edge reconstruction[2]. 3). *Global Cluster-aware Learning*. BOND utilizes DBSCAN, a structural clustering method, for paper clustering. Throughout the training process, global clustering benefits from pseudo-clustering labels derived from the local metric learning module's paper representations. In a reciprocal manner, these global clustering outcomes provide valuable cues for the local metric learning module, resulting in enhanced paper representations. This collaborative interaction substantially improves the quality of the final paper clustering results.

The primary contributions of BOND are summarized as follows:

- To the best of our knowledge, we are the first to introduce an end-to-end bootstrapping strategy for paper similarity learning and paper clustering to address the SND problem.
- BOND unifies local metric learning and global cluster-aware learning as multi-task promoting, fostering joint learning and mutual enhancement of both modules.
- Extensive experimental results highlight substantial performance gains achieved by BOND. Notably, even without intricate ensemble and post-match strategies, BOND significantly outperforms the previous Top-1 method of WhoIsWho [4]. Now, BOND currently holds the top position on the WhoIsWho leaderboard[1].

---

[2]Notably, BOND can adapt any graph model based on an attention-aggregation scheme as the base encoder.

## 2 RELATED WORK

### 2.1 Non-graph-based Methods

Non-graph-based SND methods traditionally rely on the careful definition of hand-crafted features to quantify pairwise paper similarity [6, 36]. These similarity features are typically classified into two main categories: relational features and semantic features. Relational features commonly encompass the extraction of coauthor similarity, which serves as a pivotal signal for distinguishing authors based on their social connections. On the other hand, semantic similarity features are frequently derived from various attributes such as paper titles, abstracts, keywords, and similar attributes [21], aiming to disambiguate authors by assessing the coherence of research topics between two papers. However, these approaches grapple with limitations in their ability to effectively harness the intricate higher-order structure inherent in paper similarity graphs.

### 2.2 Graph-based Methods

Graph-based SND methods construct either heterogeneous or homogeneous graphs to leverage high-order information [32, 34, 43]. With the development of network representation learning and graph neural networks, some representative methods [5, 26, 42, 45] have been integrated into the SND problem, enabling the utilization of node features and the graph structure via aggregating information from neighboring nodes. In a notable example [33], a heterogeneous graph is employed to model paper connections. A pair-wise RNN network with attention mechanisms is applied for both blocking and clustering. Another approach, proposed in [28], combines two types of graphs: a person-person graph established by connecting papers with shared coauthors and a document-document graph representing the similarity between the content of publications. These methods adhere to the relational and semantic aspects discussed in Section 2.1. However, these approaches usually conduct paper similarity learning and clustering separately, thus facing the challenge of harmonizing local distance metric learning with downstream global clustering tasks. In this work, we strive to jointly learn both local and global information within an end-to-end learning framework on multi-relational local linkage graphs.

### 2.3 Clustering Methods for SND Problem

The determination of cluster numbers is a crucial aspect of the SND problem, and it has been the subject of investigation in prior research [34, 46]. Hierarchical clustering algorithms [13, 24] operate on the premise that papers with higher similarity should be merged initially, followed by the clustering of the resulting merged clusters. A two-stage algorithm introduced in [41] leverages the clustering outcomes from the initial stage to generate clustering features for the subsequent stage. Furthermore, several methodologies have incorporated spectral clustering to enhance the efficiency of clustering procedures [11, 25]. In contrast, our model utilizes DBSCAN as the clustering strategy, which forms clusters based on density and does not necessitate predefined cluster sizes. Moreover, we seamlessly integrate the clustering algorithm into our disambiguation framework in an end-to-end manner, facilitating the joint optimization of local metric learning and global clustering.

# 3 PROBLEM DEFINITION

In this section, we present the preliminaries and the problem formulation of from-scratch name disambiguation.

*Definition 3.1.* **Paper**. A paper $p$ is associated with multiple attributes, i.e., $p = \{x_1, \cdots, x_F\}$, where $x_f \in p$ denotes the $f$-th attribute (e.g., co-authors/venues) and $F$ is the number of attributes.

*Definition 3.2.* **Author**. An author $a$ contains a paper set, i.e., $a = \{p_1, \cdots, p_n\}$, where $n$ is the number of papers authored by $a$.

*Definition 3.3.* **Candidate Papers**. Given a name denoted by $na$, $\mathcal{P}^{na} = \{p_1^{na}, \ldots, p_N^{na}\}$ is a set of candidate papers authored by individuals with the name $na$.

PROBLEM 1. ***From-scratch Name Disambiguation (SND)***. *Given candidate papers $\mathcal{P}^{na}$ associated with name $na$, SND aims at finding a function $\Phi$ to partition $\mathcal{P}^{na}$ into a set of disjoint clusters $C^{na}$, i.e.,*

$$\Phi(\mathcal{P}^{na}) \rightarrow C^{na}, \text{ where } C^{na} = \{C_1^{na}, C_2^{na}, \cdots, C_K^{na}\},$$

*where $C^{na}$ represents the resulting clusters, each cluster consists of papers from the same author, i.e., $\mathbb{I}(p_i^{na}) = \mathbb{I}(p_j^{na}), \forall(p_i^{na}, p_j^{na}) \in C_k^{na} \times C_k^{na}$, and different clusters contain papers from different authors, i.e., $\mathbb{I}(p_i^{na}) \neq \mathbb{I}(p_j^{na}), \forall(p_i^{na}, p_j^{na}) \in C_k^{na} \times C_{k'}^{na}, k \neq k'$. $\mathbb{I}(p_i^{na})$ is the author identification of the paper $p_i^{na}$.*

Notably, BOND tries to tackle the SND problem based on the built paper-author multi-relational graphs (see Section 4.1 for detailed information). Compared to the traditional methods which are based on non-graph-based methods. Recent attempts [29, 33] imply that building relational graphs can characterize the fine-grained correlations among papers and authors, thus facilitating the following SND algorithms. The experimental results also indicate the graph-based SND framework consistently outperforms other non-graph-based ones ranging from 6.0% to 32.6%.

# 4 METHODOLOGY

As previously discussed, conventional approaches typically adopt a decoupled pipeline for addressing the from-scratch name disambiguation problem. This pipeline involves initially capturing local relationships among papers and subsequently performing global clustering based on the localized information. Regrettably, this two-stage optimization process hinders the seamless diffusion of information between the two distinct task modalities, making it challenging to self-correct cumulative errors. In response to this limitation, we introduce BOND, an end-to-end approach for name disambiguation. It starts by building a multi-relational graph to capture paper relationships (Section 4.1). Then, local metric learning is performed to enhance paper representations (Section 4.2), and a clustering-aware learning algorithm is used to understand global relationships (Section 4.3). Finally, BOND optimizes both tasks together within an end-to-end algorithm (Section 4.4). The framework is illustrated in Figure 2. In the following sections, we delve into the specifics of each individual component.

## 4.1 Multi-relational Graph Construction

To estimate local relationships, i.e., pairwise similarities, among candidate papers, we create a local linkage graph, denoted as $G^{na} =$ $(\mathcal{P}^{na}, E^{na})$, for each name $na$. Here, $\mathcal{P}^{na}$ is the set of candidate papers, and $E^{na} \in \mathcal{P}^{na} \times \mathcal{P}^{na}$ represents the edge set between these papers. To ensure the preservation of comprehensive relationships while eliminating extraneous connections among papers, it is imperative to precisely specify the edges and node features.

**Edge Construction.** We measure paper similarities through multiple pathways that signify authorship, such as co-author (authored by individuals with the same name, except for the disambiguated name), co-venue (sharing the same conference or journal), and co-organization (affiliated with the same institution). While traditional approaches [9, 27, 45] have frequently relied on the co-author relationship as a primary measure of paper similarities, recent empirical research [4] has shed light on the effectiveness of alternative paper attributes in capturing semantic or structural aspects of paper similarity. In light of these findings, we opt to incorporate three distinct paper attributes—namely, co-author, co-org, and co-venue—as factors for measuring paper similarity.

We employ different linguistic word-match metrics to capture the exact and relative similarities between these paper attributes. For co-author and co-venue relationships, we use the *word overlap* metric to calculate similarities between papers. However, for co-organization relationships, where the attribute often contains redundant words, we use the *Jaccard Index* as the metric. We determine whether to add edges between papers based on thresholds determined through validation performance. Our experiments in Section 5.4 indicate that the performance is sensitive to these pre-defined thresholds.

**Node Feature Initialization.** The semantic information captured by node input features is equally essential for identifying paper authorship. Following the analysis in [4], the combination of paper titles, author organizations, and keywords proves to be crucial, thus we also adopt these paper attributes to initialize the input features. For simplicity and effectiveness, we train a Word2Vec [23] model on the WhoIsWho corpus and encode each word in the relevant paper attributes into a low-dimensional continuous vector. The superiority of Word2Vec is discussed in Section 5.6. These vectors are then summed to create paper embeddings $X_i$.

## 4.2 Local Metric Learning

In the absence of supervised authorship signals within the candidate papers, we rely on semantic and structural paper features for quantifying paper similarities. Existing approaches often employ unsupervised paper embeddings obtained through network embedding (NE) [29, 42] or graph neural networks (GNNs) [22, 29, 33]. Similarly, we employ a graph auto-encoder [17], comprising an encoder and a decoder, for the purpose of learning precise paper representations. The encoder leverages graph attention networks (GAT) due to their adaptability in learning edge weights through the attention mechanism. The paper representations are derived through the following expression,

$$H^{''} = \text{GAT}(W_e, A(\mathcal{G}), H^{'}) = g(A(\mathcal{G})H^{'} W_e^{\top} + b_e), \quad (1)$$

where $H^{'}$ represents the input paper embeddings (set to $X$ in the first layer), while $W_e$ and $b_e$ denote the projection matrix and the bias of the encoder, respectively. $A(\mathcal{G})$ represents the learned attention matrix, and $g$ is the activation function. The edge weight is

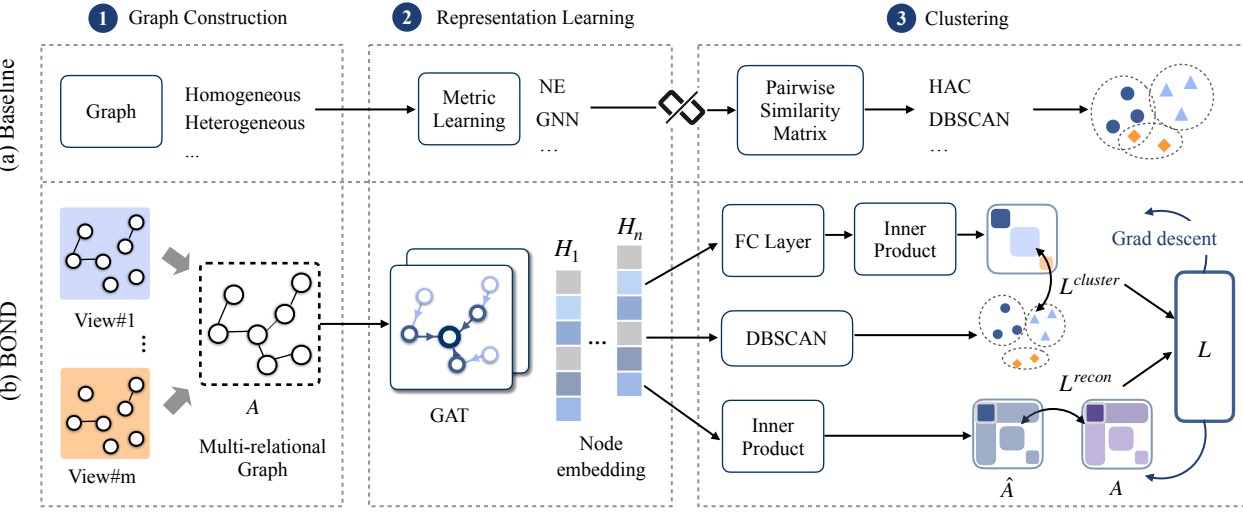

**Figure 2: The overall framework of BOND and other SND methods.** NE: network embedding; HAC: hierarchical agglomerative clustering; our proposed framework, as depicted in (b), integrates metric learning and clustering within a multi-task learning framework. By optimizing the weighted sum of reconstruction loss and cluster-aware loss, the global information derived from the clustering component can reciprocally guide the local information extracted from the reconstruction part.

parameterized as follows,

$$e_{ij} = c^{\top}([W_e H_i' || W_e H_j']), j \in N_i, \quad (2)$$

For node $i$, we calculate the coefficients between $i$ and its neighbors $j$ separately. $W_e$ is a shared parameter matrix to extend dimension and $c$ is for projecting the high-dimension to a real number. $||$ is the concatenation operator. The attention weight in $A(\mathcal{G})$ is calculated as follows,

$$\alpha_{ij} = \frac{\exp(e_{ij})}{\sum_{k \in N_i} \exp(e_{ik})}, \quad (3)$$

We utilize multi-head attention to obtain richer hidden representations and employ two GAT layers in the encoder to obtain hidden embeddings $H$.

The decoder is defined as the inner product between the hidden embeddings,

$$\hat{A} = \text{sigmoid}(H^{\top} H). \quad (4)$$

The objective function is designed to minimize the reconstruction error of the adjacency matrix through the cross-entropy loss,

$$\mathcal{L}^{\text{recon}} = \sum_{i=1}^{N} \sum_{j=1}^{N} (A_{ij} \log p(\hat{A}_{ij}) + (1 - A_{ij}) \log(1 - p(\hat{A}_{ij})), \quad (5)$$

where $A$ is the original adjacency matrix of $\mathcal{G}$ and $N$ is the node number in the graph. The local minima achieved through reconstructing the linkage among papers yields appropriate paper representations, forming the foundation for the subsequent process.

## 4.3 Global Cluster-aware Learning

In traditional methodologies, the paper embeddings denoted as $H$, which result from local linkage learning, are typically employed for estimating pairwise similarities between papers. Subsequently, these methods utilize clustering algorithms like DBSCAN to partition the papers into distinct clusters to achieve disambiguation.

However, a common oversight in these approaches is the underutilization of global clustering results, which have the potential to enhance the quality of paper representations obtained through local optimization. We posit that it is possible to effectively perform paper similarity learning and clustering in an end-to-end manner, thereby capitalizing on the mutual reinforcement of these two tasks.

To this end, we leverage DBSCAN to generate cluster labels due to its flexibility in cluster number specification, denoted as $Y$, based on the paper embeddings $H$. These labels provide essential global alignment signals. To capitalize on these signals and enhance the quality of paper representations, we introduce a fully connected layer, which processes the paper embeddings $H$ to produce output representations $C$, aiming to learn cluster-aware representations,

$$C = H W_c^{\top} + b_c, \quad (6)$$

where $W_c$ and $b_c$ represent the projection matrix and bias parameters of the fully connected layer, respectively.

Then, we attain the pairwise relationships $C$ between nodes through inner product operations, i.e., $C = CC^{\top}$. To facilitate a comparison between the global alignment label $Y$ generated by DBSCAN and the local results $C$, we also convert $Y$ into the adjacency matrix $\mathcal{Y}$,

$$\mathcal{Y} = [\mathbb{I}(Y_i = Y_j)]^{N \times N}, \quad (7)$$

where $C_{ij}$ indicates the similarity score between node $i$ and node $j$, while $\mathcal{Y}_{ij}$ signifies whether node $i$ and node $j$ belong to the same cluster label[3].

Finally, we define the cluster-aware loss using the cross-entropy objective to bootstrap the global alignment signals to the local linkage learning module,

---

[3]Here we regard nodes with label −1 as the same cluster for simplicity.

$$\mathcal{L}^{\text{cluster}} = \sum_{i=1}^{N} \sum_{j=1}^{N} \left( \mathcal{Y}_{ij} \log p(C_{ij}) + (1 - \mathcal{Y}_{ij}) \log(1 - p(C_{ij})) \right). \quad (8)$$

## 4.4 Joint Objective Optimization

In this process, we aim to find a balance between the cluster-aware loss $\mathcal{L}^{\text{cluster}}$ and the reconstruction loss $\mathcal{L}^{\text{recon}}$, which are crucial components for our BOND. We achieve this by using a weighted sum of these losses, as represented by the following equation:

$$\mathcal{L} = \lambda \mathcal{L}^{\text{cluster}} + (1 - \lambda) \mathcal{L}^{\text{recon}} \quad (9)$$

where $\lambda$ is a hyper-parameter empirically set to 0.5. We employ the clustering labels $Y$ of the last epoch as the final prediction results.

The training procedure of BOND is outlined in Algorithm 1. For each epoch, in line 2-3, we obtain hidden representation $H$ via GNN encoders and cluster-aware representation $C$ successively. In line 4, we get the outputs $\hat{A}$ and $C$ of local metric learning and cluster-aware learning, respectively. In line 5, pseudo labels $Y$ are generated based on hidden representation $H$. Finally, in line 6-8, we compute the total loss $\mathcal{L}$ based on separate loss of each task and then optimize the model via back propogation.

Local metric learning serves the purpose of enhancing the model's comprehension of paper similarities and the underlying graph topology. However, it has a vulnerability to noise, which may stem from local linkage graphs constructed based on feature similarity. In contrast, global cluster-aware learning aligns representations with the goal of the SND problem. These two tasks offer diverse perspectives and mutually enhance each other.

---

**Algorithm 1:** The Joint Objective Optimization Procedure

**Input** : Multi-relational Graph $G^{na}$, the multi-task loss $\mathcal{L}^{\text{cluster}}$, $\mathcal{L}^{\text{recon}}$ and the loss weight $\lambda$. (GD: gradient descent).

**Output** : Obtain model with parameters $\theta$.

1 **for** $iter = 1, 2, \cdots, T$ **do**
2     Get hidden representation $H$ with Eq.(1) via local metric learning.
3     Get cluster-aware representation $C$ with Eq.(6) on $H$.
4     Get reconstruction adjacency matrix $\hat{A}$ with Eq.(4) and pairwise class proximity matrix $\mathcal{C}$.
5     Get pseudo-label $Y$ with DBSCAN on $H$.
6     Compute reconstruction loss $\mathcal{L}^{\text{recon}}$ with Eq.(5) and cluster-aware loss $\mathcal{L}^{\text{cluster}}$ with Eq.(8).
7     Calculate the joint loss $\mathcal{L}$ as the weighted sum of $\mathcal{L}^{\text{recon}}$ and $\mathcal{L}^{\text{cluster}}$.
8     Update $\theta$ via GD on $\nabla_\theta \mathcal{L}$.
9 **end for**

---

## 4.5 Time Complexity

The local metric learning module adopts GAT, thus the time complexity of layer $k$ is $O\left(D_k^2 N + D_k E\right)$, where $D_k$ is the embedding size in layer $k$, $N$ is the number of nodes and $E$ is the number of edges. The global clustering module adopts DBSCAN whose average time complexity is $O(N \log N)$. The time complexity to build

the reconstruction adjacency matrix is $O\left(N^2 D_k\right)$. Since the embedding size is far smaller than the number of nodes or edges, the time complexity of BOND is $O\left(N^2 + E\right)$.

## 5 EXPERIMENTS

The source code for this work is openly accessible to the public[4].

## 5.1 Experimental Setup

**Datasets.** We utilize the WhoisWho-v3 dataset [4] as our experimental benchmark, which is the largest human-annotated name disambiguation dataset to date. This dataset comprises 480 unique author names, 12,431 authors, and 285,252 papers, each with attributes like title, keywords, abstract, authors, affiliations, venue, and publication year. Following the WhoIsWho competition, we divide it into training, validation, and testing sets in a $2 : 1 : 1$ ratio based on author names.

**Baselines.** We've conducted a rigorous comparison of our method with various SND approaches. To ensure fairness, the number of clusters has been aligned with the true value.

- **Louppe et al. [21]:** employs a classification model trained for each paper pair, aiming to determine if they are authored by the same individual. They utilize carefully designed features and semi-supervised cut-off strategies to form flat clusters of papers.
- **IUAD [19]:** constructs paper similarity graphs based on co-author relationships. It enhances the collaboration network using a probabilistic generative model that integrates network structures, research interests, and research communities.
- **G/L-Emb [46]:** utilizes common features between papers to create paper-paper networks. It learns paper representations by reconstructing these networks and employs hierarchical agglomerative clustering (HAC) for clustering.
- **LAND [30]:** constructs a knowledge graph with papers, authors, and organizations as nodes and multi-relational edges. It uses BERT [7] for initializing entity embeddings and employs the LiteralE [18] knowledge representation learning method. Then, it also uses HAC for clustering.
- **PHNet [29]:** builds a heterogeneous paper network and employs heterogeneous graph convolution networks (HGCN) for node embeddings. It uses graph-enhanced HAC for clustering, requiring a predefined cluster size.
- **SND-all [4]:** applies metapath2vec for extracting heterogeneous relational graph features along with soft semantic features. It utilizes DBSCAN for clustering and involves bagging in network embedding training. Additionally, it employs a rule-based post-match algorithm for handling outliers and cluster formation.

**Evaluation Metric** The evaluation of clustering results is based on pairwise Precision, Recall, and F1 [4, 46]. Subsequently, a macro metric is derived by averaging these performance metrics across all the individual names.

## 5.2 Main Results

In Table 1, we conduct a comprehensive comparative analysis of various author disambiguation methods. Louppe et al. distinguishes

---

[4] https://github.com/xlbhzzz/BOND

**Table 1: Results of from-scratch name disambiguation (%).**

| Model | Precision | Recall | F1 |
|---|---|---|---|
| Louppe et al. | 68.05 | 46.32 | 55.12 |
| IUAD | 58.82 | 65.22 | 61.63 |
| G/L-Emb | 50.77 | 84.64 | 63.48 |
| LAND | 61.20 | 61.12 | 61.12 |
| PHNet | 65.91 | 68.32 | 67.09 |
| SND-all- | 81.68 | 89.97 | 85.62 |
| BOND | **82.07** | **94.21** | **87.72** |

**Table 2: Improvement of unified loss (%) and the statistical significance.**

| Loss | Precision | Recall | F1 | P-value |
|---|---|---|---|---|
| Only Cluster. | 79.83 | **96.42** | 87.35 | 0.0014 |
| Only Recon. | 77.58 | 94.19 | 85.08 | 9.4640E-5 |
| Unified loss | **82.34** | 95.27 | **88.33** | / |

itself by relying on supervised pair-wise classification, underpinned by meticulously designed features. In contrast, other methodologies adopt unsupervised techniques for learning from raw data.

As an illustration, IUAD establishes coauthor networks through the mining of frequent collaborative relationships, subsequently incorporating probabilistic generative models that leverage similarity functions within the collaborative network. The relatively suboptimal performance of IUAD can be attributed to its heavy reliance on co-author relationships. In contrast, our approach considers a broader spectrum of relationships, thereby preserving comprehensive structural paper connections. Furthermore, when juxtaposed with G/L-Emb, LAND, PHNet and SND-all, each of which takes into account distinct types of connections, our model emerges as a notable frontrunner in terms of performance. G/L-Emb enhances local distance learning between papers through global semantic representations. LAND leverages knowledge embedding, while PHNet harnesses the capabilities of a heterogeneous graph neural network, and SND-all deftly integrates soft semantic features with heterogeneous relational graph features. Notably, our approach stands apart by operating as an end-to-end solution for author disambiguation, seamlessly harmonizing the twin processes of learning paper similarities and conducting clustering. This harmonious integration culminates in the generation of remarkably discriminative representations, thereby distinguishing our methodology from the decoupled approaches of our counterparts.

### 5.3 Ablation Study

In this section, we provide a justification for the effectiveness of each component within our framework.

**Effect of the different losses.** As depicted in Table 2, we compare the performance of joint loss, i.e., $\mathcal{L}$, and the use of single loss, i.e., $\mathcal{L}^{recon}$ and $\mathcal{L}^{cluster}$, on the validation set. The performance of the cluster-aware learning task surpasses the local metric learning task, suggesting that downstream clustering tasks can provide more accurate guidance for representation learning. The unified task demonstrates an improvement of +0.98% over the cluster-aware

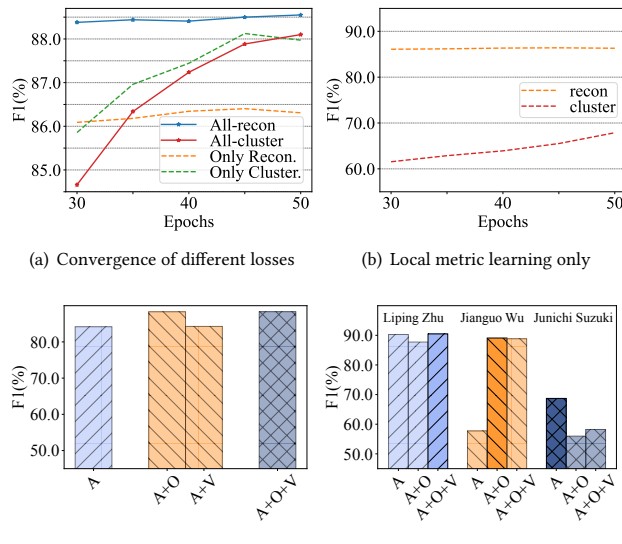

(a) Convergence of different losses

(b) Local metric learning only

(c) Multi-relational features

(d) Different name performances

**Figure 3: Effect of different losses and multi-relational features.** (a): *All*: training on all loss; *All-recon*: the clusters of local metric learning; *All-cluster*: the outputs of cluster-aware learning. (b): *recon/cluster*: the clustering results when using only the outputs of either local metric learning or cluster-aware learning; (c) and (d): *A*: CoAuthor; *O*: CoOrg; *V*: CoVenue.

learning task and +3.25% compared to the local metric learning task. These findings validate the efficacy of unifying the two tasks, as they complement and enhance one another.

We further compare the performance achieved by the unified loss with the single loss as the training goes, as illustrated in Figure 3(a). The reconstruction performance of the unified loss, i.e., the blue line, is better than the results with the model using single reconstruction loss for training, i.e., the orange line. While in Figure 3(b), take single reconstruction loss as an example, training processes in a two-stage way, causing a disconnect between metric learning and clustering. The results indicate that joint optimization can enhance the performance of both tasks, achieving superior results compared to separate single-task approaches.

**Effect of multi-relational features.** Figure 3(c) demonstrates the impact of multi-relational features. Our study of multi-view graphs is constructed in a cumulative fashion. CoA denotes the co-author relationships, excluding the author to be disambiguated, and it yields high-quality relations. CoO represents co-organization relationships of the disambiguation author. CoV refers to the co-venue relationships of the compared two papers.

The performance of Co(A+O) surpasses that of CoA by +4.13%, suggesting that co-organization contains valuable information and fills the gap that co-author cannot cover. Since co-venue relationships are not that discriminative to represent the authorship, we set the probability to 0.1 to reserve the co-venue edges. In our study, CoV doesn't take effect for the single model of BOND, but achieves clear improvements for our ensemble model when combined with CoA and CoO relational features.

However, Figure 3(d) substantiates the distinct characteristics of local linkage graphs across different names by manipulating

the multi-relational graphs employed by BOND. For example, in the case of Jianguo Wu, the incorporation of the Co-organization relation[5] results in a performance improvement of +31.37%. This finding suggests that Co-organization uncovers information that is not present in Co-author relationships. In contrast, the performance of the names Liping Zhu and Junichi Suzuki is compromised, indicating that Co-organization may introduce noise in these instances. Similarly, the Co-venue enhances performance by +2.73% in Liping Zhu and +2.23% in Junichi Suzuki. However, it either weakens or has no effect on Jianguo Wu. These results imply that the Co-author and Co-organization relationships already provide sufficient information for disambiguating these author names.

In light of these observations, our motivation is directed towards the ensemble of diverse models by employing edge-purging strategies. This approach will be elucidated in the following section.

## 5.4 WhoIsWho Competition

To assess the effectiveness of our proposed approach, we have extended the BOND to be evaluated on the widely recognized WhoIsWho benchmark[6], which has attracted the attention of over 3,000 researchers. Notably, our model, bolstered by ensemble learning techniques and the introduction of a post-match strategy (denoted as BOND+), has remarkably secured the first position in this benchmark. In the following subsection, we provide a detailed introduction to these enhanced strategies and conduct a meticulous ablation analysis to comprehensively evaluate their influence.

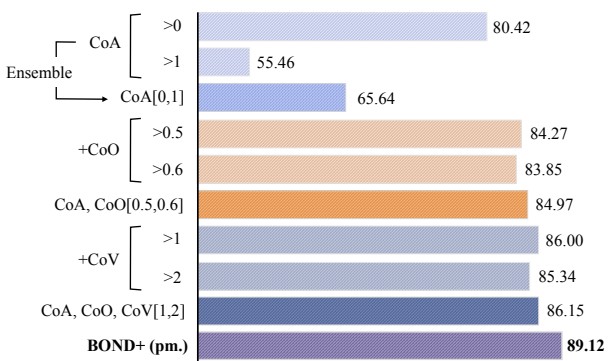

**Figure 4: The results of ablation analysis of our ensemble model. (%)** ">0" signifies that the threshold is set to 0, resulting in the removal of edges with values below 0. "CoA[0,1]" denotes the ensembling of models with coauthor values greater than 0 and 1, respectively. "CoA, CoO[0.5,0.6]" refers to the multi-view model with coauthor and coorg edges, when coauthor greater than 0 and 1 and coorg greater than 0 and 1, respectively.

**Ensemble learning.** Since different multi-relational features provide different inductive biases for name disambiguation, we argue that the ensemble of multiple models trained on local linkage graphs built with different relational features could complement each other. In this study, we train multiple models with different relational features and employ a voting mechanism for their output labels. As

---

[5]corresponding to the threshold in Figure 4
[6]http://whoiswho.biendata.xyz/

**Table 3: Performance of transductive learning and inductive learning (%)**

| Settings | Precision | Recall | F1 |
|---|---|---|---|
| Transductive | **85.18** | 94.97 | **88.55** |
| Transductive-fixed | 83.24 | **95.65** | 87.51 |
| Inductive | 84.15 | 91.49 | 86.19 |

illustrated in Figure 4, an increase in the number of models can result in a performance enhancement of up to +5.73%.

**Post-match.** Outliers generated by DBSCAN can be post-matched to either existing paper clusters or new clusters. Following the idea of WhoIsWho contest winners, we conduct similarity matching between unassigned papers (outliers) and assigned papers based on paper titles, keywords, co-authors, co-venues (CoV), and co-organizations (CoO). We adopt *tanimoto distance* to calculate CoO and CoV similarities and character matching on paper keywords and titles. If the combined similarity score exceeds a pre-defined threshold, i.e., 1.5 in our method, the papers are assigned to their respective groups. As illustrated in Table 4 , post-match improves the performance by +2.97%.

## 5.5 Transductive v.s. Inductive Learning

In this section, we scrutinize the performance of our model in both transductive and inductive scenarios. In the transductive context, we pursue the training of distinct models for each graph, which is constructed for individual names. Consequently, we adjust the dimensions of the output representations $C$ in accordance with the specific number of nodes within the given graph.

In the inductive setting, we train the model using all graphs in the training set, which are randomly shuffled in each epoch. The size of the fully connected layer $C$ is fixed. Subsequently, the model is frozen during inference on unseen graphs in the test set.

As depicted in Table 3, the transductive setting exhibits a performance improvement of +2.36% compared to the inductive setting, also with an absolute gain of 1.04% over a fixed size of $C$, indicating that the transductive setting with adaptive output size suits SND problem most. This superiority can be attributed to the transductive approach's capability to capture the unique characteristics of each graph pertaining to individual names. Additionally, the adaptability of the fully connected layers, accommodating different graph sizes, contributes to the observed performance gain.

## 5.6 Can Pre-trained Models help?

For GNN encoders, we employ Word2Vec to initialize node features. We conduct a comparative analysis between Word2Vec and other pre-trained models, including OAG-BERT [20] and SciBERT [2]. OAG-BERT is pre-trained on the corpus of Open Academic Graph [44], while SciBERT is trained based on papers in the Semantic Scholar corpus. We use the `oagbert-v2-sim` version of OAG-BERT, which is fine-tuned on WhoIsWho training corpus. As illustrated in Table 4, Word2Vec surpasses OAG-BERT by 1.84% and outperforms SciBERT by 4.05%, showing the clear gap between semantic knowledge embodied in large pre-trained models and the discriminative information required by name disambiguation task. A promising direction could be fine-tuning large language models

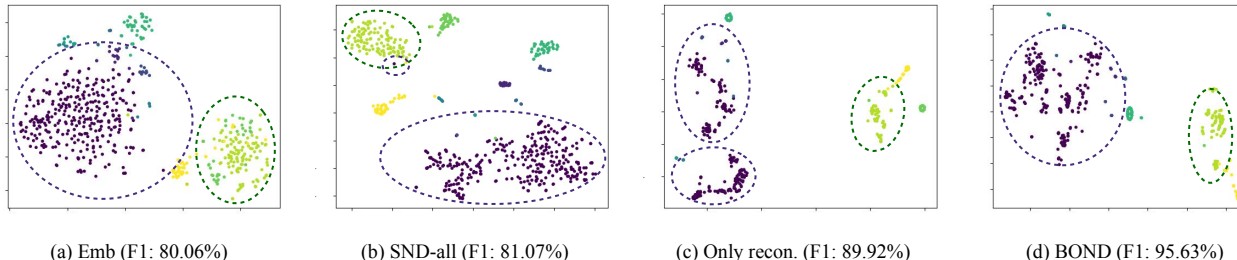

(a) Emb (F1: 80.06%)    (b) SND-all (F1: 81.07%)    (c) Only recon. (F1: 89.92%)    (d) BOND (F1: 95.63%)

**Figure 5: Visualization of paper embeddings and the F1 score on a name reference `Jianrong Li`.** Each color in (a)-(d) denotes a ground-truth cluster. "Emb" indicates the initial node features. "Only recon." represents the single usage of reconstruction loss. SND-all, "Only recon." and BOND are all initialized by "Emb".

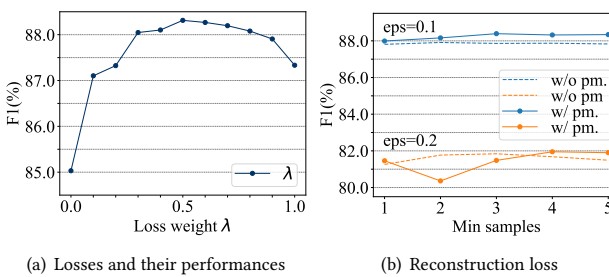

(a) Losses and their performances   (b) Reconstruction loss

**Figure 6: Analysis of hyper-parameters.**

**Table 4: Semantic Embedding Methods (%).**

| Methods | Precision | Recall | F1 |
|---------|-----------|--------|-----|
| OAG-BERT | **82.39** | 91.58 | 86.74 |
| SciBERT | 76.64 | 95.15 | 84.90 |
| Word2vec | 82.36 | **95.25** | **88.34** |

with the objectives specific to author name disambiguation, which is deferred to future work.

This observation suggests that large pre-trained models may embody substantial semantic knowledge from extensive datasets, but they exhibit noticeable bias when compared to the discriminative information required for the name disambiguation task.

### 5.7 Hyper-parameter Sensitivity

In this subsection, we investigate the performance variation when adjusting main hyper-parameters in BOND.

**Sensitivity of the weight of cluster-aware loss.** We examined how the parameter $\lambda$ impacts name disambiguation performance in the range of $[0, 1]$. The results in Figure 6(a) indicate that the best $\lambda$ value is 0.5, striking a balance between local linkage learning and cluster-aware learning. A larger $\lambda$ approaching 1 yields better results than $\lambda \rightarrow 0$, demonstrating the effectiveness of our proposed end-to-end cluster-aware learning component.

**Parameters of DBSCAN.** The maximum distance between neighboring samples *eps*, and the minimum samples in a neighborhood *min_samples*, can both impact the performance of DBSCAN, as illustrated in Figure 6(b). Our observations reveal that *eps* has a more pronounced impact on performance, and reducing it from 0.2

to 0.1 leads to a significant improvement, implying that the strict restriction of neighboring distance would generate better clustering results. The relationship between *min_samples* and post-match is intertwined. As demonstrated by the line with circle dots, performance enhances as *min_samples* increases from 1 to 5, resulting in more outliers that could be addressed by post-match strategies.

### 5.8 Visualization of Embeddings for Clustering

For a more intuitive comparison, we depict the paper embeddings trained via four methods: "Emb", SND-all, "Only recon.", and BOND. The resultant embeddings are projected into a 2-dimensional Euclidean space via t-SNE algorithm [38], as shown in Figure 5.

In these figures, each color represents the corresponding ground-truth cluster. Figure 5a shows blurred boundaries between different clusters. Figure 5b demonstrates that SND-all performs better for green (top) and yellow (left) clusters, as it considers both semantic and relational information. However, purple classes are still located in scattered areas. "Only recon." performs relatively better than SND-all, as it models structural relations between papers better via graph attention machanism. BOND demonstrates the best within-cluster compactness and clearest between-cluster boundaries, indicating the superior effectiveness of joint local metric learning and global cluster-aware learning.

### 6 CONCLUSION

In this work, we introduce the first attempt to address the from-scratch name disambiguation problem by mutually enhancing the local and global optimal signals within an end-to-end framework. Specifically, our global clustering task utilizes local pairwise similarities to create pseudo-clustering outcomes, and these global optimization signals are used as feedback to further refine the local pairwise characteristics. Our extensive experiments validate the effectiveness of each component in our proposed framework. In the future, we aim to mitigate inherent biases in different author names and explore commonalities across various names by leveraging extensive disambiguation data and large language models.

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

# A APPENDICES

## A.1 Implementation Details of BOND

In practice, our encoder is structured with two GAT layers, and the decoder employs inner product methodology for the graph auto-encoder. To identify optimal hidden layer dimensionalities, we explore a range of values from $32, 64, 128, 256, 512$. Similarly, the dimensionality of the fully-connected layer is examined within the set $32, 64, 100, 256$. For the joint objective learning, the weight parameter $\lambda$ for cluster-aware learning is fixed at 0.5. All model parameters are initialized using the Xavier uniform distribution [10] and optimized through the Adam optimizer [15]. Hyperparameters such as the learning rate and weight decay are systematically explored within the range of $1e^{-4}$ to $3e^{-3}$. Each model associated with an author's name is meticulously trained over a course of 50 epochs. All experiments are conducted on an NVIDIA GTX 3090Ti GPU.

**Table 5: Analysis of GNN Encoder (%).**

| Models | Precision | Recall | F1 |
|--------|-----------|--------|-------|
| GCN | 81.71 | 94.04 | 87.44 |
| GIN | 71.16 | **96.0** | 81.75 |
| GAT | **82.36** | 95.25 | **88.34** |

## A.2 Analysis of GNN Encoders

We employ GAT as the GNN encoder in our model. We also compare GAT with other popular GNN models, including GCN [16] and GIN [40]. As illustrated in Table 5, GAT exhibits superior performance compared to GCN and GIN, achieving 1.03% improvement over GCN and 8.06% improvement over GIN w.r.t. pairwise F1. This is attributed to GAT's ability to assign adaptive importance to different edges through its attention mechanism. Moreover, we also observe that employing heterogeneous GNNs like RGCN [31] and RGAT [3] doesn't bring clear edge over homogeneous GNNs. Thus, GAT maintains both good effectiveness and effciency in our model.

## A.3 Out-Layer Size of the Fully Connected Layer

**Table 6: Analysis of Compress ratio (%).**

| Ratio | Precision | Recall | F1 |
|-------|-----------|--------|-------|
| 0.03 | 75.67 | **95.31** | 84.36 |
| 0.1 | 78.90 | 94.87 | 86.15 |
| 0.3 | 79.30 | 95.06 | 86.47 |
| 1.0 | **82.38** | 94.24 | **87.91** |
| 3.0 | 82.12 | 92.70 | 87.09 |

In the training of the cluster-aware learning module, we utilize the transductive setting and dynamically adapt the size of the output layer within the fully connected layer based on the compression ratio multiplied by the number of nodes in the graph. As presented in Table 6, there is an observable performance enhancement of +3.55% when the compression ratio is extended from 0.03 to 1.0. This outcome highlights the module's effectiveness in capturing the unique characteristics of individual name-associated graphs while accommodating the adaptability of the fully connected layers.

