# OpenReview forum: "BOND: Bootstrapping From-Scratch Name Disambiguation with Multi-task Promoting"
_ACM.org/TheWebConf/2024/Conference — TheWebConf24_

### Official Review · Reviewer_SB5f · 2023-11-25

**Novelty:** 5
**Technical Quality:** 5

**Review:**

Summary:
This paper focuses on the problem of information exchange in the commonly used pipeline-based approaches for the From-Scratch Name Disambiguation (SND) task. The paper introduces an end-to-end model that synergistically leverages local and global informative signals to mutually reinforce each other within the proposed framework.

Strengths:
1. The experimental design of this study is comprehensive and addresses potential concerns of the readers. The paper conducts various experiments that effectively cover several aspects of interest, making the findings highly persuasive.
2. By integrating local and global signals within an end-to-end framework, the paper provides a novel perspective that effectively addresses the problem at hand.

Weaknesses:
1. The clustering algorithm may struggle in large datasets - not a reject reason. It would be better to illustrate the efficiency of the method.
2. There are some typos in the paper. For example, "SND-all-" in Table 1 seems to be "SND-all". It would be better to carefully correct these typos.

**Questions:**

1. The reported main experimental results are different from the online leaderboard. According to the website http://whoiswho.biendata.xyz/#/, the authors' model achieved an F1 score of 0.89719, while SND-all attained an F1 score of 0.89216. However, Table 1 of the paper reports lower results. It would be better to clarify the detailed differences.
2. The paper adopts GAT and reconstructs the adjacency matrix to learn local information. However, in my mind, such a constrain can also learn global information. So it would be better to discuss more on the motivation to use GAT to capture local information.
3. The paper leverages global information by making nodes more similar when they are assigned to the same cluster. However, I think if the two nodes are incorrectly assigned to the same cluster, the model may encourage the two nodes to be more similar. It would be better to discuss more on this case.

**Reviewer Confidence:**

3: The reviewer is confident but not certain that the evaluation is correct

**Scope:**

3: The work is somewhat relevant to the Web and to the track, and is of narrow interest to a sub-community

---

### Official Review · Reviewer_JzXP · 2023-11-28

**Novelty:** 6
**Technical Quality:** 6

**Review:**

The paper has merits. It addressed the fromscratch name disambiguation problem by mutually enhancing the local and global optimal signals in an end-to-end framework (BOND) . That is, the global clustering task utilizes local pairwise similarities to create pseudo-clustering outcomes, and these global optimization signals are used as feedback to further refine the local pairwise characteristics.
Extensive experiments have been done to proposed BOND’s efficacy. Besides, the paper is well written and easy to follow.

I have following suggestion for the paper.
(1) Some technical details need more explanation, e.g., in equation (4), why the decoder is defined as the inner product between the hidden embeddings.

**Questions:**

My question:

(1) In Table 2, recall with Only cluster loss is higher (96.42) is greater than the recall with unified loss. Any reason for that ?

**Reviewer Confidence:**

4: The reviewer is certain that the evaluation is correct and very familiar with the relevant literature

**Scope:**

4: The work is relevant to the Web and to the track, and is of broad interest to the community

---

### Official Review · Reviewer_ktUE · 2023-11-28

**Novelty:** 4
**Technical Quality:** 5

**Review:**

The research presented in this paper advances the field of Name Disambiguation through an innovative end-to-end approach. Distinct from the conventional two-step method of establishing local relationships between papers before global clustering, this approach leverages graph attention networks to derive node embeddings from a Multi-relational Graph. These embeddings are then used for cluster learning, optimized through a unified objective function. The method's efficacy is validated by its first-place finish in the WhoIsWho Challenge.

Advantages:
- Utilizes a Multi-relational Graph to capture the local relations between papers effectively.
- Employs joint objective optimization to harmonize the model's capabilities in reconstruction and clustering tasks.

Limitations:

- The explanation of edge construction, specifically the metrics of word overlap and the Jaccard Index, lacks clarity. The author should provide a comprehensive description of these metrics and validate their effectiveness through ablation studies.
- The paper mentions the sensitivity of predetermined thresholds in line 316 but does not offer adequate discussion or experimental justification for how to choose these thresholds.
- Figure 5 limits its demonstration to the embedding of a single sample, raising concerns about the method's generalizability. A more extensive comparison across multiple samples or introducing holistic metrics for embedding assessment would further enhance the study's credibility.

**Questions:**

- This work contains FNN, DBSCAN, and Inner Product Methods for Global clustering learning. Could you evaluate every single method's performance in the ablation study part?
- The topic and method of this work are closely related to this paper: "Author disambiguation by hierarchical agglomerative clustering with adaptive stopping criterion." I suggest that you discuss it relative to your framework. There may be others similar to this work.

**Reviewer Confidence:**

4: The reviewer is certain that the evaluation is correct and very familiar with the relevant literature

**Scope:**

4: The work is relevant to the Web and to the track, and is of broad interest to the community

---

### Official Review · Reviewer_c5ez · 2023-12-05

**Novelty:** 4
**Technical Quality:** 5

**Review:**

The paper presents a new framework named BOND for the from-scratch name disambiguation task which aims to cluster documents authored by individuals with identical names into distinct groups. The authors claim existing methods split this task into two separate tasks – document similarities estimation and document clustering, which hinders effective information exchange and thus their end-to-end solution BOND can better synergize local and global signals. Specifically, it utilizes local pairwise document similarities to guide global clustering, which in turn generates pseudo-clustering labels and these global signals are then used to enhance local pairwise characterizations. On a benchmark dataset, authors conduct experiments to show BOND’s superiority over other methods and enhance it with ensemble & post-match techniques to win the WhoIsWho competition.

Overall the paper is clearly written and the studied problem is interesting (although the presented scope is a little bit narrow). The pipeline is overall reasonable and the experiment settings are complete.

There are a few concerns about this paper. First, I feel the technical novelty is somewhat limited. If I understand correctly, the paper simply treats the DBSCAN outputs as silver labels and defines the L_cluster loss by using them as ground truth. What happens if the DBSCAN outputs in the first iteration are very bad? This scenario might not happen for the WhoIsWho benchmark as its academic domain has relatively clean attributes (e.g., authorship, venues, etc) but in many real name disambiguation problems, the initial DBSCAN clustering outputs can be noisy and I am wondering if the current iterative optimization framework will propagate this noise. Second, for the experiments, I feel the authors need to compare with those joint embedding + clustering work (something of the favor like [1], not exactly this work, but those methods with the clustering and embedding are jointly optimized). For ablation studies, I sugges author testing more clustering algorithms (e.g., HDBSCAN, AP clustering, non-parametric density methods, etc) to see if the BOND framework can generalize. Finally, one small typo, in Table 1 “SND-All-” with one additional dash.

[1]. Unsupervised Deep Embedding for Clustering Analysis, https://proceedings.mlr.press/v48/xieb16.pdf

**Questions:**

1. Have you tried to replace the DBSCAN algorithm to other clustering algorithms in the BOND framework?
2. Have you tried to replace the multi-relational graph embedding algorithm to other direct HIN-based embedding algorithm?

**Reviewer Confidence:**

2: The reviewer is willing to defend the evaluation, but it is likely that the reviewer did not understand parts of the paper

**Scope:**

3: The work is somewhat relevant to the Web and to the track, and is of narrow interest to a sub-community

---

### Decision · Program_Chairs · 2024-01-22

**Decision:**

Accept

**Comment:**

The paper develops an end-to-end framework called BOND for name disambiguation by mutually enhancing the local and global optimal signals. The global clustering task utilizes local pairwise similarities to create pseudo-clustering outcomes, while these global optimization signals are used as feedback to further refine the local pairwise characteristics. Empirical results show that the proposed method has promises.